**Subject Category:**
Biology (whole organism)

behaviour/biophysics/evolution

collective behaviour, animal movement, flocculation

**Author for correspondence:**
Nigel R. Franks
e-mail: nigel.franks@bristol.ac.uk

# Social flocculation in plant–animal worms

Alan Worley[1], Ana B. Sendova-Franks[2]
and Nigel R. Franks[1]

[1]School of Biological Sciences, University of Bristol, 24 Tyndall Avenue, Bristol BS8 1TQ, UK
[2]Department of Engineering Design and Mathematics, UWE Bristol, Frenchay Campus, Coldharbour Lane, Bristol BS16 1QY, UK

 AW, 0000-0002-7734-7841; ABS-F, 0000-0001-9300-6986;
NRF, 0000-0001-8139-9604

Individual animals can often move more safely or more efficiently as members of a group. This can be as simple as safety in numbers or as sophisticated as aerodynamic or hydrodynamic cooperation. Here, we show that individual plant–animal worms (*Symsagittifera roscoffensis*) can move to safety more quickly through flocculation. Flocs form in response to turbulence that might otherwise carry these beach-dwelling worms out to sea. They allow the worms to descend much more quickly to the safety of the substrate than single worms could swim. Descent speed increases with floc size such that larger flocs can catch up with smaller ones and engulf them to become even larger and faster. To our knowledge, this is the first demonstration of social flocculation in a wild, multicellular organism. It is also remarkable that such effective flocculation occurs where the components are comparatively large multicellular organisms organized as entangled ensembles.

## 1. Introduction

Social behaviour can bring tremendous advantages, including safety in numbers [1] and the ease and efficiency of coordinated movement [2,3] when groups of organisms move through a resistive medium [4]. Examples include chevron formations adopted by birds in flight [5], queuing of spiny lobsters [6], swimming of newborn dolphins [7] and alignment in schools of fish [8]. Moving collectively, even bacteria achieve significantly greater swimming speeds compared with isolated individuals [9]. Here, we show the hydrodynamic advantages gained by plant–animal worms from their use of flocculation (figure 1).

The marine acoel flat worm *Symsagittifera roscoffensis* (Ludwig Von Graaf 1891) is found in intertidal regions of a number of sandy beaches in France, the Channel Islands, the UK and Portugal. One special feature of the adult worm is that it derives all of its energy from photosynthesis occurring in its symbiotic algae (*Platymonas convolutae*). The worms are visible (during the

**Figure 1.** Flocculation in plant–animal worms as seen from above. (*a*) Immediately after the end of agitation, when the worms are hooked together into dark flocs. (*b*) 40 s later, when they have separated and reverted to their usual behaviour. For more details, see electronic supplementary material, section S1.

day and when the tide is out) as dark green masses in shallow pools bathed by run-off water. At night and when invaded by the tide, they burrow into the sand [10,11].

Adult individuals are approximately 1.7 mm long and typically swim, horizontally, at approximately 1.8 mm s$^{-1}$ [12]. They exhibit several social behaviours: they interact with each other more frequently than would occur by chance; they swim in polarized groupings; and at higher concentrations, they readily form circular mills [12,13]. Here we show a further example of their collective behaviour. If a container of S. roscoffensis in water is suddenly disturbed, the worms agglomerate and their colour apparently becomes darker, a result of the opacity of the dense groupings (figure 1; electronic supplementary material, section S1, video S1, figures S1 and S2). The groups drop rapidly to the bottom of the container. We hypothesize that this aggregation is an aid to a swift descent and, despite the worm size being two orders of magnitude higher than the largest particles involved in flocculation, we have interpreted our results in terms of this process.

A rapid reaction by the worms may be required whenever an unpredictable wave breaks over a group; without this response, the worms could be washed into deep water. When the worms encounter the bottom of a sample bottle, they disperse and their green colour becomes apparent once more (figure 1*b*); we suggest that the rapid dispersion is necessary to enable individuals to burrow into the sand.

This provisional interpretation raises several questions—can such groupings of these worms really descend more quickly than the more straightforward strategy of swimming vertically? How are the worms organized within flocs? And can we recognize any of the established hydrodynamic relations between floc size and terminal velocity in their behaviour?

Flocculation is the coagulation of particles in suspension to form masses variously described as fleecy or clumpy. It may be deliberately induced, such as in water treatment [14,15] or brewing [16], as a source of microalgae for biofuels [17]—or natural, as in the formation of marine algal (diatom) flocs [18]. The first study of flocculation as a cooperative behaviour was in yeast [19]. If the flocculation process involves the trapping of gas bubbles, it can result in buoyancy, but otherwise, the process aids sedimentation—aggregation enables the particles to descend more rapidly through the medium.

Our hypotheses are grounded in the physics of flocculation, which we will introduce briefly here. A worm of length $L = 1.7$ mm swimming horizontally at $v \sim 1.8$ mm s$^{-1}$ has a Reynolds number $Re \sim 3$ [20]. So, if large numbers of worms are involved as a floc, and if the descent speed exceeds the normal horizontal swimming speed, the Reynolds number will exceed the limit ($Re \sim 1$) up to which Stokes' Law applies. The drag equation for a small sphere moving through a viscous fluid is also known as Stokes' drag. For illustrative purposes, however, and on the assumptions (i) that the clusters may be taken as spherical, (ii) that Stokes' Law applies, (iii) that the density of settling objects is small enough that the behaviour of a cluster is unaffected by interactions with other particles, and (iv) that the presence of container walls may be neglected, the terminal velocity $v$ emerges from the balance between the gravitational force moderated by buoyancy $F_m = (4/3)\pi a^3(\rho_m - \rho_f)g$ and the hydrodynamic drag force $F_h = 6\pi\eta av$, where $a$ is the composite particle radius, $\rho_m$ its density, $\rho_f$

the fluid density, $g$ the acceleration due to gravity and $\eta$ the dynamic viscosity of the fluid. Replacing the particle radius $a$ by its characteristic dimension $L = 2a$, we have the familiar relation [21]:

$$v = \frac{L^2(\rho_m - \rho_f)g}{18\eta},$$ (1.1)

If we further assume that the floc density is unchanging with size:

$$v \propto L^2$$ (1.2)

On the basis of equation (1.2), and as $L$ increases, falling under gravity should at some stage be more effective than descending as an individual swimmer. More directly, if worms form a cluster and descend purely under gravity, it is advantageous to do so as a large cluster. These considerations provide a useful foundation for further analyses even though experiments with real and inanimate flocs demonstrate that they do not conform strictly to the above model. Rather than assembling one particle at a time, for example, larger flocs form from the adhesion of smaller ones. This cluster–cluster flocculation [22] produces structures which are more open, with densities that decrease with increasing size and with increasing amounts of interstitial water [23]. This added complexity may help with the problem of extending hydrodynamic theory of moving flocs into the $Re > 1$ regime because of fluid flow through the porous floc interior [24]. Though the Reynolds number is still elevated, the porosity has the effect of reducing the turbulent wake and allows calculations to proceed using Stokes' Law, albeit with a different constant of proportionality.

In a separate approach, flocs are treated as self-similar fractal entities [25] for which the relationship between the number of particles $n$, each of size $L_p$ in a three-dimensional floc of size $L$ is taken as:

$$n = \left(\frac{L}{L_p}\right)^D,$$ (1.3)

and the variation of settling speed with $L$ is:

$$v \propto L^{D-1},$$ (1.4)

where $D$ is the fractal dimension, capacity dimension or Hausdorff dimension of the assembly. $D$ can range in value from 3.0, representing perfect packing with no interstitial space, to 1.0, which arises when all the particles are arranged in a straight line. Recognizing that $D$ may not be the same for small, compact flocs and larger more diffuse ones [26] produces a more general expression for the terminal velocity for fractal flocs which is recognizably a refinement of equation (1.1):

$$v = \frac{K L_p^{3-D(L)}}{L^{1-D(L)}},$$ (1.5)

where $K = (\rho_m - \rho_f)g/18\eta$ and $D(L)$ represents the variability of the fractal dimension $D$. For flocs assembled from 1.0 µm particles, equation (1.5) is in good agreement with grouped data used in a meta-analysis [27].

Equation (1.5) is based upon Stokes' Law and thus strictly only applies for Reynolds number $< \sim 1$. Nevertheless, measurements on waste-activated sludge flocs [24] carried out with floc sizes in the range 150 µm to 10 mm and for Reynolds numbers in the range 0.03–80 demonstrate terminal velocities (SI units):

$$v \sim 1.17\, L^{0.99}.$$ (1.6)

Although equations (1.5) and (1.6) arise from the flocculation of particles significantly smaller than our worms, the empirical power-law form exemplified by equation (1.6) and the more detailed equation (1.5), based on flocs with a fractal structure and a varying dimensionality $D$, will be the basis for a comparison with our results.

# 2. Material and methods

## 2.1. Experimental set-up and procedure

The work was undertaken on Guernsey's northern shore between 10 and 14 June 2017. Worms were collected daily and held in sunny conditions until required. The apparatus for studying flocculation

in a column of seawater is shown in electronic supplementary material, figure S3. We used a 500 mm clear acrylic tube of external diameter 16 mm and internal diameter 12 mm, mounted vertically and connected to a horizontal axle such that 170 mm of the tube was below the axis of rotation. For each run, we pipetted worms with a little seawater into a clear glass vial (inner dimensions $75 \times 16.8$ mm diameter) which was then fitted to the bottom of the acrylic tube with $15 \times 0.5$ mm O-rings (internal and cross-sectional diameter, electronic supplementary material, figure S3*b*). We then filled the tube slowly from the top with filtered seawater (passed through a plastic sieve with square holes of side 0.85 mm) and capped it in such a way as to trap a substantial air bubble, height of $17.2 \pm 8.0$ mm (mean $\pm$ s.d.), $n = 16$, at the top (electronic supplementary material, section S2 and figure S4). The total length of the resulting fluid column was 555 mm. After a short period for the worms to settle, we rotated the acrylic tube quickly through $180°$, causing the air bubble to rise through the tube and deliver a large disturbance to the worms, now at the top, which were thereby induced to begin their emergency descent. A Canon Sureshot G16 camera recorded a portion of the tube from a distance of 15 cm to produce a video with a resolution of $1080 \times 1920$ pixels at 30 fps over 2 min from the moment just before the rotation began. When the tube has been inverted, the image area extends from 330 to 415 mm below the top of the fluid column. Other features of the run were captured with a hand-held Canon G7 camera recording video with $768 \times 1024$ pixels at 15 fps. At the end of the 2 min run, we flushed the tube and vial and photographed the worms in a shallow tray. By counting the number of worms in a known fraction of the tray area, the number of worms used was estimated at $3700 \pm 1100$ (mean $\pm$ s.d, $n = 28$). The tube and vial were cleaned and the procedure repeated with a different sample for the 17 runs (electronic supplementary material, table S1). All worms were returned to the same beach after the experiment.

## 2.2. Analysis

For each floc found, the frame numbers of its image entry and exit times were noted and the $y$-coordinate of its position determined for each intermediate image. The images were cropped in the $x$-direction to feature only the 16 mm acrylic tube and in the $y$-direction by $p$ pixels above and below the $y$-value; $p$ varies with floc size and falls in the range 40–120 pixels. The selected area was then extracted and converted to a binary image, minor features deleted and an ellipse fitted to the floc with major axis $L$, minor axis $h$, using the dimensions of the tube as calibration. Our measure of the characteristic floc size is taken as the mean of the values of $L$. The terminal velocity $v$ was determined directly from the entry and exit times in the image.

## 2.3. Composition of smallest flocs

This estimate was derived as follows. The fitted ellipses were interpreted as oblate spheroids with volume $V = \pi L^2 h/6$. This gave each of the two smallest flocs a volume $V \sim 0.21$ mm$^3$ (flocs #16 and 17; electronic supplementary material, table S1), which was divided by the typical volume for a single worm of 0.04 mm$^3$ and multiplied by an assumed packing fraction of 0.5 with the resulting estimate of 2–3 worms.

## 2.4. Concentration of single worms

These estimates were obtained at 1 s intervals for the whole of each run. As before, each image was cropped in the $x$-direction to include only the water column. After background detection and conversion to a binary image with a black background, the light pixels, representing worms, were counted. The procedure is imperfect—overlapping worms will be under-represented, for example—but the aim was to determine the mean vertical speed of the single worms rather than their absolute number. For comparison, a Gaussian distribution of speeds was generated and, for each time after the arrival of the bubble, the maximum and minimum worm speeds which could feature in the image were calculated. The integral under the Gaussian between these two speeds represents the worms currently in the image and so can be compared with the observed pixel count. The mean, standard deviation and peak height of the Gaussian were then varied to obtain the best fit over the whole of the distribution (electronic supplementary material, section S3 and figure S5).

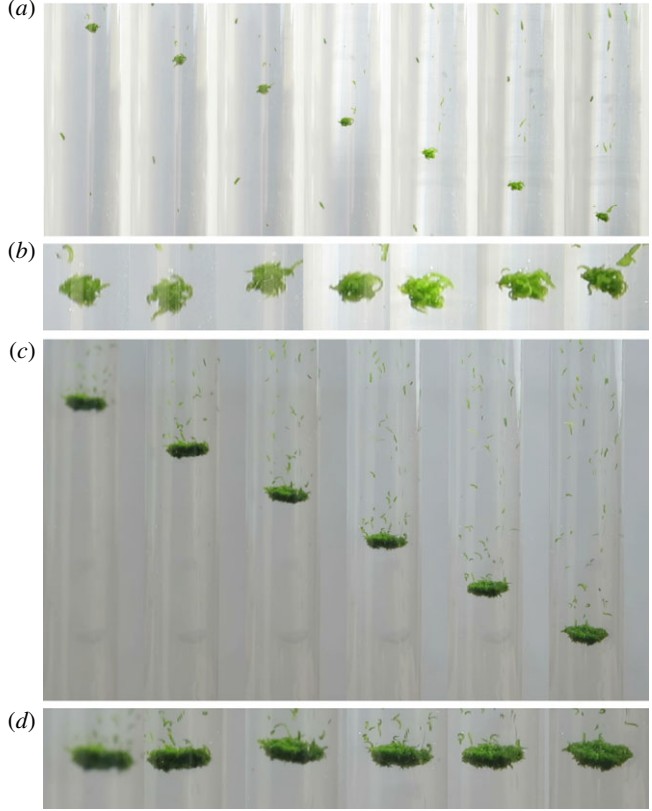

**Figure 2.** Two flocs descending in a column of seawater. (*a*) Successive images of floc #1 at 0.5 s intervals. The distance covered by the floc increases linearly with time and the terminal velocity is readily measured. A solitary worm appears ahead of the floc, but this worm is moving more slowly, as are several identifiable worms following it. Above the floc in the last few frames, the large following group of single worms is seen entering the field of view. (*b*) Close-up images of floc #1. The mean floc diameter $L = 2.12$ mm. The floc has a relatively open structure and is slightly flattened in its direction of travel. At the top right of the last frame, it has just shed a pair of worms which initially appeared connected and were later seen to separate. (*c*) Successive images of floc #2 at 0.5 s intervals. The floc appears to have a toroidal structure (see also electronic supplementary material, figure S7). It is descending into clear water and shedding many single worms that follow above it. Such worms may be seen in successive images and are slower than the floc itself. (*d*) Close-up images of floc #2. The mean floc diameter $L = 8.97$ mm.

## 2.5. Curling up following disturbance

To examine the possible mechanism by which flocs might form, we performed the following experiment. We introduced a small number of worms in ∼0.5 ml seawater into a $12.5 \times 12.5 \times 45$ mm cuvette (Neolab E-1641) and recorded 50 fps 1080p MP4 video with a Sony RX100 Mk V camera, beginning just before we rapidly introduced a further 2.5 ml of seawater from a pipette. We examined the images and identified the image numbers at which the injected water first arrived at the bottom of the cuvette, the last image in which normal worms were observed and the first image in which most of the worms were curled up.

# 3. Results

## 3.1. Floc formation

We performed a total of 36 runs in which we filmed flocculation in a column of seawater. Seventeen of these runs yielded a total of 22 flocs that reached the recording area (see Material and methods). The runs that did not yield filmable flocs may have been associated with smaller bubbles that did not cause sufficient disturbance (electronic supplementary material, section S2). We observed two predominant behaviours of flocs: amalgamation, as faster-moving flocs overtook slower ones, and reduction in size by the evaporation of worms (figure 2).

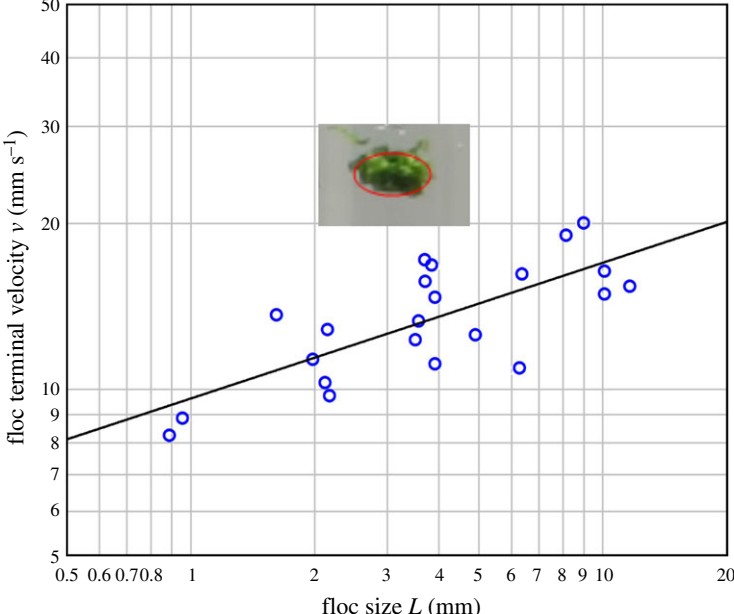

**Figure 3.** Terminal velocity $v$ versus floc size $L$. $L$ was measured as the mean major axis of the fitted ellipse (see Material and methods). For further details, see electronic supplementary material, table S1. (*inset*) floc #12 with its fitted ellipse.

## 3.2. Floc size and composition

The floc characteristic dimension $L$, as measured by the mean major axis of the fitted ellipse, ranged between 0.89 and 11.61 mm (electronic supplementary material, table S1). The smallest flocs consisted of only 2–3 worms (see Material and methods). At the other extreme, the largest flocs probably contained two orders of magnitude more worms than the smallest (figure 2$d$). There was a change in shape with size; larger flocs were flatter and had better-defined boundaries. The progression was from a closely knit cluster of very few worms, to an oblate spheroid with rough boundaries (figure 2$b$), to a thick pancake and finally a torus (figure 2$d$). The floc aspect ratio between the mean minor and mean major axes of the fitted ellipse, $h/L$, decays exponentially with floc size (electronic supplementary material, figure S6).

## 3.3. Floc terminal velocity

Terminal velocity increases sub-linearly with increasing floc size (figure 3). A power law of the form $v = 9.63\,L^{0.247}$ (exponent: 99% CI $0.247 \pm 0.142$, $t = 4.97$, $p < 0.001$) fits the data well ($R^2 = 55.25\%$, Anderson–Darling test for normality of residuals: $AD = 0.426$, d.f. $= 22$, $p = 0.287$).

## 3.4. Single worms

All flocs were falling more quickly than both the 1.8 mm s$^{-1}$ mean horizontal speed of a single worm [12] and the mean vertical speed of single worms within the same run (electronic supplementary material, table S1, section S3, and figure S5). Most flocs, particularly the larger ones, also exceeded the speeds of at least 97.5% of the single worms in the same run (figure 4). In fact, individual worms within flocs achieve an approximately 50% more rapid descent than even the fastest individual worms and with very little energy expenditure (figure 4).

## 3.5. Curling up following disturbance

The worms reacted rapidly to the disturbance; in the separate test in which water was squirted into a cuvette containing worms and initially only very little water, worms curled up within a quarter of a second of the arrival of the injection (figure 5 and table 1). The mean time over the 10 runs was 0.24 s (s.d. $= 0.05$ s).

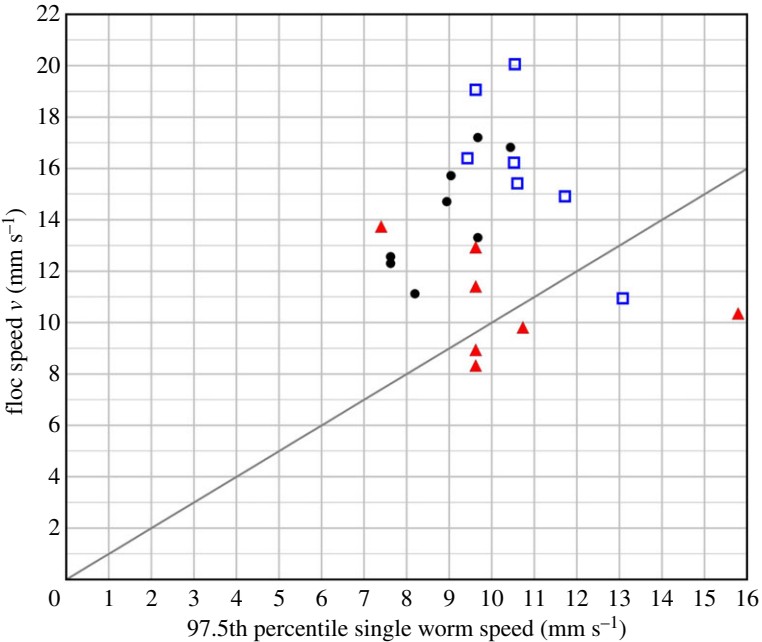

**Figure 4.** The floc terminal velocity plotted against the 97.5th percentile for the speed of single worms, moving outside flocs, in the same run. The line marks the equality of the two speeds. The flocs are sorted into three classes: ▲ small ($n = 7$), ● medium ($n = 8$), ☐ large ($n = 7$).

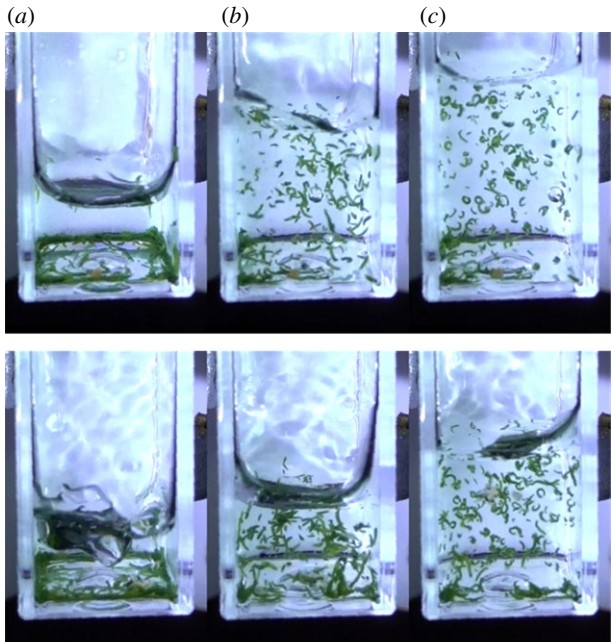

**Figure 5.** Reaction to disturbance. Two out of the 10 replicates showing (a) the arrival of water at the base of the cuvette, (b) the last image with mainly normal worms and (c) the first image in which most of the worms are curled up. Some worms are already entangled with others. The water continues to arrive after the time of the third image.

## 4. Discussion

The flocs in the present study feature a living organism which can modify its body shape, and to the best of our knowledge, the larger toroidal structures we report have not been observed in conventional flocs. In addition, the range of floc sizes covers little more than a single decade. These factors make it difficult

**Table 1.** Reaction to disturbance. The image numbers, i1, i2 and i3, correspond to the stages shown in figure 5a−c for each of the 10 replicates. The final column gives the time interval between the arrival of the water and the first image in which most of the worms are curled up (that is between i3 and i1).

| replicate | image number | | | $\Delta t$ (s) |
|---|---|---|---|---|
| | i1 | i2 | i3 | |
| 1 | 163 | 169 | 172 | 0.18 |
| 2 | 101 | 110 | 114 | 0.26 |
| 3 | 47 | 53 | 59 | 0.24 |
| 4 | 36 | 47 | 54 | 0.36 |
| 5 | 49 | 56 | 61 | 0.24 |
| 6 | 76 | 83 | 89 | 0.26 |
| 7 | 102 | 107 | 112 | 0.20 |
| 8 | 70 | 76 | 81 | 0.22 |
| 9 | 72 | 76 | 81 | 0.18 |
| 10 | 67 | 76 | 80 | 0.26 |

to identify any trend in figure 3 other than a straightforward power law such as equation (1.6), though the slope we have determined is smaller. Potentially, our results could also be accommodated within equation (1.5), which passes through a peak.

We have followed our flocs through the last 10 cm of a 30 cm descent from the top of the column. This mimics a significant wave disturbance compared with the 0−3 cm deep pools in which the worms are normally found. Some replicates did not yield filmable flocs and we think this might be associated with a lack of sufficient perturbation. This suggests that the tendency to floc in individual worms might be tuneable by natural selection. The flocs that were successfully filmed maintained their integrity over large distances. Nevertheless, all the large flocs were observed to shed worms. This is consistent with the observation, for inanimate flocs, that floc strength decreases with increasing size [28]. We note also that our observation of hollow flocs is consistent with the pressure on the surface being greatest at the bottom centre of the descending mass. It is, however, tempting to speculate that there may be another reason for the stability of the smaller flocs. S. roscoffensis feature a statocyst [10], a hollow sphere containing a freely moving chalk ball which acts to advise on the direction of gravity. It is likely to be the disturbance detected in this organ which promotes the emergency response leading to the worms aggregating and falling. As we have shown, when the disturbance ends, the worms immediately begin to disengage (figure 1; electronic supplementary material, figures S1 and S2). It seems possible, therefore, that the calm descent of the larger flocs may provide a signal that the emergency is over. Smaller flocs, however, are less stable and may tumble [29], depending on their aspect ratio and Reynolds number. Such tumbling may provide continuous stimulation to the statocysts of the worms and cause them to prolong their attachment. It is tempting to speculate that aggregations of few worms need to be tightly bound on purely geometric grounds, and that their tumbling ensures that this is the case. When greater numbers are involved, worms can appear normal and still be sufficiently convoluted to remain as a floc. Clearly, further work is needed on the morphology of flocs, the body shapes and behaviour of the worms within them, and the processes which take place as flocs merge.

We have shown that flocs descend, typically, approximately 50% more rapidly than single worms. If worms, exposed on a beach, are overwhelmed by the unpredictable arrival of a large wave as the tide approaches, this could result in their being borne out, irreversibly, into deep water; their ability to react to such an event may be crucial for their survival. The 50% speed advantage we find for worms in flocs is significant; performance differences as small as 5% are claimed [30] to confer a useful evolutionary advantage.

Our preliminary work on the possible mechanism behind floc formation in these worms shows that they react to a simulated wave by curling up within a quarter of a second. By curling up, the worms can coil around one another to form flocs. Hence, flocs are based on a rapid behavioural response on the part of the worms and as such are an example of social behaviour.

# 5. Conclusion

We have shown that isolated worms agglomerate into flocs and as a result can reach the comparative safety of the substrate more quickly than a free-swimming single worm. In nature, the worms are often not isolated but occur as biofilms [12]. Hence, they would be well placed, in the event of being overwhelmed by an unexpected wave, to form the entangled ensembles [31] we have observed—so flocs can be the result of collective behaviour, and as such are a fascinating and novel example of safety in numbers.

Ethics. *Research ethics*: no research ethics assessment was required because we did not use humans or human tissue in this study. *Animal ethics*: At the time of the study, our understanding was that no ethical approval was needed for this type of research on a non-endangered and locally super-abundant worm. However, we followed the ARRIVE guidelines (https://www.nc3rs.org.uk/arrive-guidelines) and the *Guidelines for the treatment of animals in behavioural research and teaching* (https://royalsociety.org/~/media/journals/ethics/abguidelines2017.pdf?la=en-GB). The work was carried out on a publicly accessible beach and the worms and seawater used in the experiments were returned to the exact location from where they had been collected before the next tide when the worms bury themselves in the sand.

Permission to carry out fieldwork. Fieldwork with this species does not require a permit in Guernsey.

Data accessibility. Datasets supporting this article are included with the electronic supplementary material.

Authors' contributions. A.W. first recognized the possibility of flocculation in *Symsagittifera roscoffensis*. N.R.F. steered the research strategy. A.W., N.R.F. and A.B.S.-F. designed and implemented the experiments. N.R.F. and A.W. built the apparatus. A.W. analysed the photographs, videos and the resulting data. A.B.S.-F. contributed the statistical analysis. All authors discussed ideas, interpreted the results and wrote the manuscript.

Competing interests. We have no competing interests.

Funding. This research was self-funded.

Acknowledgements. We thank our departments for their support.

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
