## [Reviewer comments · Royal Society Open Science]

Review History

RSOS-181626.R0 (Original submission)

Review form: Reviewer 1

Is the manuscript scientifically sound in its present form?

Yes

Are the interpretations and conclusions justified by the results?

No

Is the language acceptable?

Yes

Is it clear how to access all supporting data?

Yes

Do you have any ethical concerns with this paper?

No

Have you any concerns about statistical analyses in this paper?

No

Recommendation?

Accept with minor revision (please list in comments)

Comments to the Author(s)

The manuscript by Worley et al demonstrated that the plant-animal worm *Symsagittifera roscoffensis* could contract and curve under disturbance which facilitates flocculation. Groups of worms flocculate so that they could descend faster than single worms to reach to bottom of shallow water to avoid being carried away by tide. The research is interesting and would complement the development of the *S. roscoffensis* in other subjects.

1. Line 34 and line 292: 'cooperative behavior' and 'social behavior' should be changed to 'collective behavior'. The horizontal circular behavior demonstrated in an earlier publication has not been shown to link to this flocculation behavior.
2. Line 23 and line 288: the aspect of energy saving was not demonstrated in the manuscript.
3. Table 1: The image number could be moved into supplementary data, keeping only $\Delta t(i3-i1)$.
4. Legend of Figure 4, the color of medium and large was the same. Different marker shape could be used for better separation.

Review form: Reviewer 2

Is the manuscript scientifically sound in its present form?

Yes

Are the interpretations and conclusions justified by the results?

Yes

Is the language acceptable?

Yes

Is it clear how to access all supporting data?

Yes

Do you have any ethical concerns with this paper?

Yes

Have you any concerns about statistical analyses in this paper?

No

Recommendation?

Accept with minor revision (please list in comments)

Comments to the Author(s)

OVERVIEW

This study covers a very interesting topic and the authors appear to have addressed the previous referee comments satisfactorily in revising the manuscript and responding to individual

comments. I have no major comments, but offer a few minor suggestions for improvement of the manuscript, and my thoughts regarding their responses to reviewer comments.

MINOR COMMENTS

Numbers and their units should always be separated by a space (e.g. 15 mm, not 15mm).

l.129 – A fuller description should be provided by what is meant by “filtered seawater” (e.g. how it was filtered, what mesh size was used etc.). Also, words like “substantial” are open to interpretation, so I would suggest a more specific description of the size of this air bubble.

l.183-184 – Does this mean that the worms aggregated in less than 50 % of the runs (i.e. 17 out of 36)? If so, then I think this is an important result that should be elaborated upon in the discussion, given the important role this behaviour potentially plays in survival the authors are discussing in relation to the evolutionary advantage of this behaviour.

l.214: Delete the comma after “above it”.

l.247: Why do the authors refer to “Our flocs”? I suggest that this use of language is too personal.

RESPONSES TO REFEREE 2 COMMENTS:

1) I agree with the authors that there is no need to increase the size of the experimental data set. If it was straightforward to carry out repeat experiments perhaps it would be of interest to do this. However, the patterns observed already and statistical analysis seems sufficiently robust as it stands, also given the nature of the fieldwork, its seasonality, and the need for self-funding, it would seem unnecessary to insist on further experiments.

RESPONSES TO REFEREE 1 COMMENTS:

2) The referee says that the “evidence for social behavior is preliminary. The rules of interaction are not fully addressed.”. This may be true, but I don’t see why this should prevent the current description of this flocculation behaviour being published in its current state.

3) I agree with the authors, that there is sufficient indication of social behaviour to leave this phrase within the current manuscript, and rely on future studies to more thoroughly dissect the nature and controls of this behaviour in these worms.

4) I agree with the authors, and do not find any inconsistencies regarding annotation for velocity.

Decision letter (RSOS-181626.R0)

04-Jan-2019

Dear Dr Franks,

The editors assigned to your paper ("Social Flocculation in Plant-Animal Worms") have now received comments from reviewers. We would like you to revise your paper in accordance with the referee and Associate Editor suggestions which can be found below (not including

confidential reports to the Editor). Please note this decision does not guarantee eventual acceptance.

Please submit a copy of your revised paper before 27-Jan-2019. Please note that the revision deadline will expire at 00.00am on this date. If we do not hear from you within this time then it will be assumed that the paper has been withdrawn. In exceptional circumstances, extensions may be possible if agreed with the Editorial Office in advance. We do not allow multiple rounds of revision so we urge you to make every effort to fully address all of the comments at this stage. If deemed necessary by the Editors, your manuscript will be sent back to one or more of the original reviewers for assessment. If the original reviewers are not available, we may invite new reviewers.

- Data accessibility

<http://datadryad.org/submit?journalID=RSOS&manu=RSOS-181626>

- Competing interests

- Authors' contributions

- Acknowledgements

- Funding statement

on behalf of Prof Kevin Padian (Subject Editor)
openscience@royalsociety.org

Associate Editor's comments:

Thank you for submitting this extension to your earlier work in Royal Society Open Science. The reviewers broadly favour publication, but have queried your response to the absence of ethical approval being sought before conducting the research. While the work is carried out on invertebrates, it is nevertheless common to seek approval from your institution's ethical committee prior to conducting interventions such as this research (see, for instance, <https://royalsociety.org/~media/journals/ethics/abguidelines2017.pdf?la=en-GB>).

The Editors would like the authors to provide an explanation of why they did not seek approvals from their ethics committee (even if none were subsequently required, it is best practice to ask the question)? We will look forward to receiving your response to this - and the the other reviewers' - query.

Reviewers' Comments to Author:

Reviewer: 1

Comments to the Author(s)

The manuscript by Worley et al demonstrated that the plant-animal worm *Symsagittifera roscoffensis* could contract and curve under disturbance which facilitates flocculation. Groups of worms flocculate so that they could descend faster than single worms to reach to bottom of shallow water to avoid being carried away by tide. The research is interesting and would complement the development of the *S. roscoffensis* in other subjects.

1. Line 34 and line 292: 'cooperative behavior' and 'social behavior' should be changed to 'collective behavior'. The horizontal circular behavior demonstrated in an earlier publication has not been shown to link to this flocculation behavior.
2. Line 23 and line 288: the aspect of energy saving was not demonstrated in the manuscript.
3. Table 1: The image number could be moved into supplementary data, keeping only $\Delta t(i3-i1)$.
4. Legend of Figure 4, the color of medium and large was the same. Different marker shape could be used for better separation.

Reviewer: 2

Comments to the Author(s)

OVERVIEW

This study covers a very interesting topic and the authors appear to have addressed the previous referee comments satisfactorily in revising the manuscript and responding to individual comments. I have no major comments, but offer a few minor suggestions for improvement of the manuscript, and my thoughts regarding their responses to reviewer comments.

MINOR COMMENTS

Numbers and their units should always be separated by a space (e.g. 15 mm, not 15mm).

l.129 - A fuller description should be provided by what is meant by "filtered seawater" (e.g. how it was filtered, what mesh size was used etc.). Also, words like "substantial" are open to interpretation, so I would suggest a more specific description of the size of this air bubble.

l.183-184 - Does this mean that the worms aggregated in less than 50 % of the runs (i.e. 17 out of 36)? If so, then I think this is an important result that should be elaborated upon in the discussion, given the important role this behaviour potentially plays in survival the authors are discussing in relation to the evolutionary advantage of this behaviour.

l.214: Delete the comma after "above it".

l.247: Why do the authors refer to "Our flocs" ? I suggest that this use of language is too personal.

RESPONSES TO REFEREE 2 COMMENTS:

1) I agree with the authors that there is no need to increase the size of the experimental data set. If it was straightforward to carry out repeat experiments perhaps it would be of interest to do this.

However, the patterns observed already and statistical analysis seems sufficiently robust as it stands, also given the nature of the fieldwork, its seasonality, and the need for self-funding, it would seem unnecessary to insist on further experiments.

RESPONSES TO REFEREE 1 COMMENTS:

2) The referee says that the “evidence for social behavior is preliminary. The rules of interaction are not fully addressed.”. This may be true, but I don’t see why this should prevent the current description of this flocculation behaviour being published in its current state.

3) I agree with the authors, that there is sufficient indication of social behaviour to leave this phrase within the current manuscript, and rely on future studies to more thoroughly dissect the nature and controls of this behaviour in these worms.

4) I agree with the authors, and do not find any inconsistencies regarding annotation for velocity.

Author's Response to Decision Letter for (RSOS-181626.R0)

See Appendix A.

RSOS-181626.R1 (Revision)

Review form: Reviewer 1

Is the manuscript scientifically sound in its present form?

Yes

Are the interpretations and conclusions justified by the results?

Yes

Is the language acceptable?

Yes

Is it clear how to access all supporting data?

Yes

Do you have any ethical concerns with this paper?

No

Have you any concerns about statistical analyses in this paper?

No

Recommendation?

Accept as is

Comments to the Author(s)

The authors have addressed my comments.

Review form: Reviewer 2

Is the manuscript scientifically sound in its present form?

Yes

Are the interpretations and conclusions justified by the results?

Yes

Is the language acceptable?

Yes

Is it clear how to access all supporting data?

Yes

Do you have any ethical concerns with this paper?

Yes

Have you any concerns about statistical analyses in this paper?

No

Recommendation?

Accept as is

Comments to the Author(s)

The authors have made a thorough job of this manuscript revision and I am happy with all the revisions made and responses to reviewer comments.

Decision letter (RSOS-181626.R1)

25-Feb-2019

Dear Dr Franks,

I am pleased to inform you that your manuscript entitled "Social Flocculation in Plant-Animal Worms" is now accepted for publication in Royal Society Open Science.

Royal Society Open Science operates under a continuous publication model (<http://bit.ly/cpFAQ>). Your article will be published straight into the next open issue and this

will be the final version of the paper. As such, it can be cited immediately by other researchers. As the issue version of your paper will be the only version to be published I would advise you to check your proofs thoroughly as changes cannot be made once the paper is published.

on behalf of Professor Kevin Padian (Subject Editor)
openscience@royalsociety.org

Reviewer comments to Author:

Reviewer: 1

Comments to the Author(s)
The authors have addressed my comments.

Reviewer: 2

Comments to the Author(s)
The authors have made a thorough job of this manuscript revision and I am happy with all the revisions made and responses to reviewer comments.

Appendix A

MS ID: RSOS-181626: *Social Flocculation in Plant-Animal Worms* by Alan Worley, Ana B. Sendova-Franks and Nigel R. Franks

Responses to Editor's and Referees' Comments

Associate Editor's comments:

- (1) Thank you for submitting this extension to your earlier work in Royal Society Open Science. The reviewers broadly favour publication, but have queried your response to the absence of ethical approval being sought before conducting the research. While the work is carried out on invertebrates, it is nevertheless common to seek approval from your institution's ethical committee prior to conducting interventions such as this research (see, for instance, <https://royalsociety.org/~media/journals/ethics/abguidelines2017.pdf?la=en-GB>).

Thank you for the positive comments about our paper. Please see under (2) below our response to the query regarding ethics in research involving animals.

- (2) The Editors would like the authors to provide an explanation of why they did not seek approvals from their ethics committee (even if none were subsequently required, it is best practice to ask the question)? We will look forward to receiving your response to this - and the the other reviewers' - query.

At the time of the study our understanding was that no ethical approval was needed for this type of research on a non-endangered and locally super-abundant worm. Having said that, we were very conscious of the ethics of research involving animals and followed the ARRIVE guidelines (<https://www.nc3rs.org.uk/arrive-guidelines>) as well as the *Guidelines for the treatment of animals in behavioural research and teaching* (<https://royalsociety.org/~media/journals/ethics/abguidelines2017.pdf?la=en-GB>) published by the journal of *Animal Behaviour* (indeed one of us served as an executive editor of this journal until recently).

More specifically, we minimised any harm to the individual plant-animal worms by returning them and the original sea water to the exact location from where they had been collected immediately after the experiment on the day of their collection. The worms bury themselves in the sand at the next tide. We minimised any disturbance to their habitat by collecting worms from a publicly accessible beach where fishermen, boat-users and walkers tread. Our collections did not involve any digging of the sand or any other disturbance of the natural state of the habitat. The numbers of worms involved in the experiments was very small in comparison with the millions in the thriving groups on the beach where we have visited them for the last seven years.

We have updated our Animal ethics statement accordingly (ll. 300-304).

Reviewer: 1

- (3) The manuscript by Worley et al demonstrated that the plant-animal worm *Symsagittifera roscoffensis* could contract and curve under disturbance which facilitates flocculation. Groups of worms flocculate so that they could descend faster than single worms to reach to bottom of shallow water to avoid being carried away by tide. The research is interesting and would complement the development of the *S. roscoffensis* in other subjects.

Thank you for these encouraging words.

- (4) 1. line 34 and line 292: 'cooperative behavior' and 'social behavior' should be changed to 'collective behavior'. The horizontal circular behavior demonstrated in an earlier publication has not been shown to link to this flocculation behavior.

We have made both these changes (ll. 34, 296).

- (5) 2. Line 23 and line 288: the aspect of energy saving was not demonstrated in the manuscript.

We have deleted "energy-saving" (l. 23) and "and with less energy expenditure" (l. 293).

- (6) 3. Table 1: The image number could be moved into supplementary data, keeping only $\Delta t(i3-i1)$.

We have considered this suggestion carefully but with all due respect we disagree because we are concerned this would sever the link between Fig. 5 and Table 1. In addition, we do not think it would save any space.

- (7) 4. Legend of Figure 4, the color of medium and large was the same. Different marker shape could be used for better separation.

We have used a blue open square for the large flocs so that both the shape and colour of the symbol are different for each floc size (Fig. 4, l. 230).

Reviewer: 2

OVERVIEW

- (8) This study covers a very interesting topic and the authors appear to have addressed the previous referee comments satisfactorily in revising the manuscript and responding to individual comments. I have no major comments, but offer a few minor suggestions for

improvement of the manuscript, and my thoughts regarding their responses to reviewer comments.

Thank you for the encouragement.

MINOR COMMENTS

- (9) Numbers and their units should always be separated by a space (e.g. 15 mm, not 15mm).

This has been implemented throughout the manuscript and ESM.

- (10) l.129 – A fuller description should be provided by what is meant by “filtered seawater” (e.g. how it was filtered, what mesh size was used etc.). Also, words like “substantial” are open to interpretation, so I would suggest a more specific description of the size of this air bubble.

After “filtered seawater”, we added “(passed through a plastic sieve with square holes of side 0.85 mm)” (ll. 128-129).

We had stated the air bubble size in the ESM but now we have added it to the main text as “height of 17.2 ± 8.0 mm (mean \pm sd), n=16” (ll. 129-130).

- (11) l.183-184 – Does this mean that the worms aggregated in less than 50 % of the runs (i.e. 17 out of 36)? If so, then I think this is an important result that should be elaborated upon in the discussion, given the important role this behaviour potentially plays in survival the authors are discussing in relation to the evolutionary advantage of this behaviour.

No, while it is true that 17 out of 36 runs yielded filmable flocs, the runs that did not yield filmable flocs may have been associated with smaller bubbles that did not cause sufficient disturbance. We have added a sentence to this effect in the Results (ll. 186-187). We have also added two sentences to the Discussion on the implications for the evolutionary advantage of floc behaviour: “Some replicates did not yield filmable flocs and we think this might be associated with a lack of sufficient perturbation. This suggests that the tendency to floc in individual worms is tuneable by natural selection.” (ll. 257-259).

- (12) l.214: Delete the comma after “above it”.

We edited the initial sentence “The floc appears to have a toroidal structure (see also figure S8), is descending into clear water and above it, are many single worms it is shedding.” Now it is replaced by “The floc appears to have a toroidal structure (see also figure S8). It is descending into clear water and shedding many single worms that follow above it.” (ll. 214-215).

(13) l.247: Why do the authors refer to “Our flocs”? I suggest that this use of language is too personal.

We replaced “Our flocs” with “The flocs in the present study” (l. 248).

RESPONSES TO REFEREE 2 COMMENTS:

(14) 1) I agree with the authors that there is no need to increase the size of the experimental data set. If it was straightforward to carry out repeat experiments perhaps it would be of interest to do this. However, the patterns observed already and statistical analysis seems sufficiently robust as it stands, also given the nature of the fieldwork, its seasonality, and the need for self-funding, it would seem unnecessary to insist on further experiments.

Thank you.

RESPONSES TO REFEREE 1 COMMENTS:

(15) 2) The referee says that the “evidence for social behavior is preliminary. The rules of interaction are not fully addressed.”. This may be true, but I don’t see why this should prevent the current description of this flocculation behaviour being published in its current state.

Thank you.

(16) 3) I agree with the authors, that there is sufficient indication of social behaviour to leave this phrase within the current manuscript, and rely on future studies to more thoroughly dissect the nature and controls of this behaviour in these worms.

Thank you.

(17) 4) I agree with the authors, and do not find any inconsistencies regarding annotation for velocity.

Thank you. We avoided the possibility for confusion between v for velocity and ν (nu) for kinematic viscosity. We deleted the latter and employed a method of calculating the Reynolds number (Re) suggested by R. McNeill Alexander in his book on *Functional design in fishes* (ll. 63-64).